# Genetic Research Progress: Heat Tolerance in Rice

**DOI:** 10.3390/ijms24087140

**Published:** 2023-04-12

**Authors:** Huaqing Liu, Bohong Zeng, Jialiang Zhao, Song Yan, Jianlin Wan, Zhibin Cao

**Affiliations:** 1Rice National Engineering Research Center (Nanchang), Jiangxi Academy of Agricultural Sciences, Nanchang 330200, China; 2Jiangxi Research and Development Center of Super Rice, Nanchang 330200, China

**Keywords:** rice, heat tolerance, QTL, molecular mechanism, molecular breeding

## Abstract

Heat stress (HS) caused by high-temperature weather seriously threatens international food security. Indeed, as an important food crop in the world, the yield and quality of rice are frequently affected by HS. Therefore, clarifying the molecular mechanism of heat tolerance and cultivating heat-tolerant rice varieties is urgent. Here, we summarized the identified quantitative trait loci (Quantitative Trait Loci, QTL) and cloned rice heat tolerance genes in recent years. We described the plasma membrane (PM) response mechanisms, protein homeostasis, reactive oxygen species (ROS) accumulation, and photosynthesis under HS in rice. We also explained some regulatory mechanisms related to heat tolerance genes. Taken together, we put forward ways to improve heat tolerance in rice, thereby providing new ideas and insights for future research.

## 1. Introduction

Climate change has affected the growth and development of major crops in the world and posed a great threat to agricultural production [1,2,3]. Climate change has increased the frequency of heat damage events that affect crop yields [4,5]. The global average temperature is expected to rise by 2–3 °C in the next 30–50 years [6]. For every 1 °C increase in the global average temperature, the average yield loss of major food crops may be as high as 7.4% [1]. In extremely high-temperature environments, HS can interfere with the normal growth, development, and metabolism of plants, and even lead to plant death in severe cases [7,8].

Rice (*Oryza sativa* L.) is the main food crop for nearly half of the world’s population [9]. Heat damage caused by climate warming is one of the main abiotic stresses affecting rice production. HS leads to a serious loss of rice yield, a decrease in quality, and a harvest index. During the reproductive growth period, including the booting, heading, and grain filling stages, rice is very sensitive to high temperatures. HS hinders the flowering and fertilization process of rice, thus reducing the seed-setting rate and yielding [10,11]. when the rice young panicles have experienced the average temperature (>33 °C) at the meiosis stage, the flower organ and pollen would not develop normally, resulting in a decrease in the seed-setting rate and abnormal floret development [12]. The high temperature (>35 °C) for 5 days at the heading and anthesis stage would affect the pollen tube elongation and normal pollen dispersal, resulting in the formation of empty and unfertilized grains [13]. The grain-filling stage is the key period of rice yield and quality formation. The high temperature at the grain filling stage can accelerate the grain filling rate and shorten the grain filling time, which not only affects the grain weight but also affects the quality [14]. It is estimated that for every 1 °C increase in the daily maximum and minimum temperature, rice yield will decrease by 10% [15]. However, the widely used rice varieties are generally sensitive to high temperatures, and the strategies to deal with HS by improving production techniques and management measures are no longer enough to maintain yield. Therefore, it is urgent to cultivate new high-temperature tolerant rice varieties [16,17].

Understanding the genetic mechanism of high temperatures in rice and cultivating heat-tolerant varieties are essential to cope with future global warming [18,19]. Identifying heat-tolerant genes, obtaining breeding intermediate materials, and cultivating heat-tolerant varieties are effective measures for rice to cope with HS [20]. Heat tolerance is a very complex quantitative trait in rice, which is regulated by multiple genes [21]. By studying the physical and chemical properties and agronomic characteristics of rice at different stages under HS, some progress has been made in the molecular genetics of high-temperature tolerance in rice, including the mapping of some QTL of high-temperature tolerance in rice. However, due to the relatively low additive effect of heat-resistant QTL, the introduction of too little QTL into a variety may not be enough to increase its heat tolerance [18,19]. Therefore, mapping, verification, and fine mapping of more major QTL as well as the design of functional SNP chips with QTL linkage markers will help to speed up the selection and integration of multiple QTLs, and then improve the efficiency of rice heat tolerance breeding. In addition, cloning the genes involved in the regulation of HS in rice is also an important direction in the application of molecular genetics to high-temperature tolerance in rice. Therefore, screening and the identification of rice germplasm resources, mining more genes, and cultivating new rice varieties with stronger heat tolerance are the main ways to alleviate HS and stabilize grain yield. In this paper, the QTL identification of rice heat tolerance, the cloning of functional genes, and the research progress of molecular mechanisms were reviewed, and the ways to improve rice heat tolerance were discussed to provide a theoretical reference for the development of rice heat tolerance molecular breeding.

## 2. QTL Identification of Heat Tolerance in Rice

The heat tolerance of rice is often deemed to be a quantitative trait, and the underlying complex mechanism is not well known. In recent years, with the continuous development of molecular marker technology and researchers’ attention to the heat damage of rice, many heat-tolerant QTLs have been located on 12 chromosomes of rice. Rice is extremely sensitive to high temperatures, and HS has effects on all stages of rice growth and development, especially at the booting stage and flowering stage. Most of the heat-tolerant QTLs published in recent years are QTLs related to the booting stage and flowering stage, and a few are related to QTL at the seedling stage (Appendix A). Common wild rice (*Oryza rufipogon* Griff.) is an important germplasm resource for rice improvement [21]. At the same time, common wild rice is also an important high-temperature tolerant rice germplasm resource, and the genetic population constructed with common wild rice as a parent is an excellent QTL mapping population for heat tolerance [22]. Cao et al. conducted QTL analysis on the interspecific near-isogenic line (NIL) constructed with Yuanjiang common wild rice Hehuatang 3 as the donor parent and Shuhui 527 as the recurrent parent. Taking the rice seed setting rate as the heat tolerance index, a heat tolerant QTL *qHTH5* at the heading and flowering stage was detected on the short arm of chromosome 5, and QTL was located within about 304.2 kb by displacement mapping [23]. After that, the BC_5_F_3_ population was constructed by crossing Yuanjiang common wild rice Hehuatang 4 with Tianfeng B, and the near-isogenic line YJ01-201 containing the target fragment of wild rice was backcrossed with Tianfeng B to construct a separate population. The QTL*qHTH10* of heat tolerance at the heading stage was located between RM25216 and RM25228 and the physical map distance was 277.1 kb. Indeed, the two generations explained the 7.4% and 15.2% phenotypic variations, respectively [24]. Using the advanced interspecific backcross introgression line IL01-15 derived from Yuanjiang common wild rice Hehuatang 4 and cultivated rice R53 as materials, Cao et al. established a secondary population and also mapped a heat tolerance QTL *qHTB1-1* at the booting stage on chromosome 1. Through homozygous recombinant screening from larger BC_6_F_2_ and BC_6_F_3_ populations, *qHTB1-1* was finely positioned in the 47.1 kb area between the markers RM11633 and RM11642. The synergism of these QTLs came from Yuanjiang common wild rice [25].

In addition to common wild rice, N22, Giza178, and Habataki are also important heat-tolerant germplasm materials for constructing heat-tolerant QTL mapping populations in rice. Ye et al. determined the fertility of the progenies of rice IR64/N22 hybrid combinations BC_1_F_1_ and F_2_ at the flowering stage and four single nucleotide polymorphisms were found in the BC_1_F_1_ population as well as four QTL related to heat tolerance in the F_2_ population. Among them, two major QTL, *qHTSF1.1* and *qHTSF4.1*, are located on chromosomes 1 and 4, thus explaining the 12.6% and 17.6% variations in the spikelet fertility under high temperatures, respectively. The tolerance allele of *qHTSF1.1* comes from IR64 and the tolerance allele of *qHTSF4.1* comes from N22 [26]. To further fine map and verify *qHTSF4*, Ye et al. developed an SNP marker based on PCR and genotyped BC_2_F_2_, BC_3_F_2_, BC_3_F_3_, and BC_5_F_2_ populations from the same hybrid combination. The interval of *qHTSF4* was reduced to about 1.2 MB. *qHTSF4* improved spikelet fertility at high temperatures at the flowering stage [19]. PS et al. mapped five thermostable QTL on chromosomes 3, 5, 9, and 12 using the N22/IR64 RIL mapping population and the 5KSNP genotyping chip. Indeed, *qSTIPSS9.1*, on the 394 kb region of chromosome 9, was an important new QTL and located a known thermostable QTL, *qSTIY5.1*, to a 331 Kb region on chromosome 5, which was involved in the interaction of two genes [27]. Kilasi et al. used an RIL population constructed by N22 and IR64 to carry out HS treatment at the seedling stage, and a total of 15 QTL related to heat tolerance were detected, of which *rlht5.1* was the main QTL for controlling the root length of HS, with an effect of 20.4% [20]. Ye et al. identified heat-resistant QTL at the flowering stage using two parental F_2_ populations and a compound hybrid F_2_ population constructed by Giza178. Four QTL were detected in the IR64/Giza178 population, two QTL were detected in the Milyang23/Giza178 population, and five QTL were detected in the IR64//Milyang23/Giza178 hybrid population, in which *qHTSF1.2*, *qHTSF4.1*, and *qHTSF6.1* overlap with the QTL in the parent population. The tolerance alleles of these QTL are derived from Giza178 except *qHTSF3.1* [18]. Zhao et al. established a set of chromosome segment substitution lines (CSSL) using Habataki and Sasanishiki as materials. Based on the analysis of three high-temperature environments, a total of 11 QTLs were identified: 8 of the 11 QTLs overlapped with the reported QTL and 3 new QTLs included *qPSLht4.1*, *qPSLht7*, and *qPSLht10.2* [28]. Zhu et al. detected heat-resistant QTL of CSSL from Habataki and Sasanishiki in two field environments. A total of 12 QTLs were detected: 5 QTLs were detected in two environments and a major QTL *qHTB3-3* was detected on the long arm of chromosome 3. After verification, *qHTB3-3* was located between RM3525 and 3-M95 with a physical distance of about 2.8 MB [12].

In addition to the above materials, Gan-Xiang-Nuo, Liaoyan 241, M9962, Nagdong, and Huang Huazhan also were used as the genetic material for mapping heat-tolerant QTL in rice. Liu et al. identified the single-segment chromosome substitution lines derived from a heat-tolerant variety, Gan-Xiang-Nuo N, and a heat-sensitive variety, Hua-Jing-Xian-74 (HJX), by QTL, and four heat-resistant QTL were detected at the flowering stage [29]. Based on the RIL mapping population, Li et al. constructed a cross between IAPAR-9 with the heat-tolerant variety Liaoyan 241 and a total of 11 heat-resistant QTLs were located. Among them, four major QTL (*qNS1*, *qNS4*, *qNS6*, and *qRRS1)* were stably detected in different environments for two years and all related to epistasis and the environmental interaction. *qNS4*, *qHTS4*, and *qRRS4* located in the RM471-RM177 region of chromosome 4 all participate in epistatic and environmental interactions and lead to phenotypic variation, indicating that this region is an important QTL locus [30]. Nubankoh et al. used the QTL-seq method to quickly locate QTL controlling spikelet fertility in the F_2_ population of heat-tolerant line M9962 and heat-sensitive variety Sinlek. One QTL was detected on chromosomes 1, 2, and 3, which were *qSF1*, *qSF2*, and *qSF3*, respectively, and four possible candidate genes was predicted [10]. Park et al. used the double-haploid line of Cheongcheong/Nagdong to analyze the QTL of high-temperature tolerance at the booting stage and mapped 19 QTL. The 2.2cM region of RM3709-RM11694 on chromosome 1 was common among the traits detected, and several candidate genes for high-temperature tolerance were mapped in this interval [31]. Chen et al. mapped the high-temperature QTL of rice at the flowering stage by using the F_2:3_ population constructed by the cross between the heat-tolerant variety Huang Huazhan and the heat-sensitive variety 9311 and also mapped a major QTL, namely *qHTT8*, on chromosome 8, located on 3,555,000–4,520,000 bp [32]. Hu et al. used relative spikelet fertility as an index of heat tolerance and carried out genome-wide association analysis on the characters of 173 rice materials under two environments. Five significantly related QTLs were detected, of which a new QTL, *qRSF9.2*, was located and its region narrowed to the 22,110,508–22,187,677 bp region on chromosome 9 [33]. There are similar heat-tolerant QTLs in rice, indicating that the metabolic pathway of heat tolerance among different rice varieties may be conservative, and some effective QTLs can be stably expressed. However, some heat-resistant QTLs were not detected together, which may be related to the different genetic backgrounds of different varieties or the differences in environmental conditions in different experiments.

## 3. The Cloning of Functional Genes Related to Heat Tolerance in Rice

The ontology system of the National Rice Data Center (https://www.ricedata.cn/, accessed on 1 February 2023) was used to query the genes related to “heat tolerance (TO:0000259)”. So far, a total of 97 genes related to heat tolerance in rice have been identified. The functional genes related to heat tolerance in rice cloned in recent years are shown in Table 1. These genes are mainly cloned by reverse genetics and mainly encode proteases.

### 3.1. Functional Genes That Play a Positive Regulatory Role

Almost all the rice heat tolerance genes cloned by forward genetics play a positive role in regulating the heat tolerance of rice. Li et al. analyzed the heat tolerance of CSSL with CG14 as the donor parent and WYJ as the recurrent parent. A major gene *TT1* was identified in the 12.69 kb region of chromosome 3 which encodes the 26S proteasome α2 subunit that is involved in the degradation of ubiquitinated proteins. Compared with *OsTT1*, *OgTT1* protects cells from HS by eliminating cytotoxic denatured proteins more effectively and by maintaining the heat response process more effectively. Overexpression of *OgTT1* in rice could significantly improve heat tolerance [34]. Shen et al. isolated and cloned a receptor-like kinase gene *ER* (*ERECTA*) from Arabidopsis thaliana. The overexpression of *ER* in rice endowed heat tolerance independent of water loss and increased biomass [35]. Wei et al. analyzed the inheritance of F_1_ and F_2_ generations of HT54 and HT13 and found that the heat tolerance of HT54 was controlled by a dominant major locus *OsHTAS* located in the 420 kb interval between InDel5 and RM7364 on rice chromosome 9 [36]. *OsHTAS* encodes a ubiquitin ligase which enhances heat tolerance by regulating stomatal closure induced by hydrogen peroxide in leaves [37]. Wang et al. used the F_2_ population obtained from the cross between *togr1-1* and Zhonghua 11 to map *TOGR1* to the 28.5 kb region on chromosome 3. *TOGR1* encodes a DEAD box RNA helicase located in the nucleolus, which is a prerequisite for rice growth tolerance [38]. Takehara et al. made a fine mapping of QTL *Apq1* which controls the appearance quality of brown rice and reduced *Apq1* to the region of 19.4 kb. The pathogenic gene of *Apq1* was *Sus3* and the increased expression of *Sus3* under high temperatures at the mature stage led to the tolerance of rice to high temperatures [39]. Xu et al. mapped *SLG1* in the 310 kb region between the markers C4 and C6 on the long arm of chromosome 12 using the F_2_ population constructed by the hybridization of *slg1* and KD8. *SLG1* encodes rice cytoplasmic tRNA2-thiolation 2 (RCTU2), which plays a key role in the response of rice seedlings and growth stages to HS [40]. Chen et al. mapped *HTS1* in the 80.2 kb region between yp430 and yp698 on the long arm of chromosome 4 using the F_2_ population produced by the hybridization between *hts1* and 9311. *HTS1* encodes a β-ketoacyl carrier protein reductase located in the thylakoid membrane, which is involved in the de novo synthesis of fatty acids and is very important for the heat tolerance of rice [41]. Zhang et al. identified a new gene locus for heat tolerance in rice, *TT3*, which is composed of two genes: *TT3.1* and *TT3.2*. Under HS, the PM located E3 ligase TT3.1 translocated to the endosome and the *TT3.1* ubiquitin chloroplast precursor protein *TT3.2* is used for vacuole degradation, thus maintaining chloroplast stability and enhancing the heat tolerance of rice [42]. Cao et al. mapped and cloned a stable heat-tolerant QTL, *qHTH5*, and *HTH5* encoding a mitochondrial pyridoxal phosphate homeostasis protein (PLPHP) on chromosome 5. The accumulation of reactive oxygen species under HS was reduced by increasing heat-induced pyridoxal-5′-phosphate (PLP) content. The overexpression of *HTH5* could increase the seed-setting rate of rice plants under HS at the heading stage. It has great potential value in improving the heat tolerance of rice to the threat of global warming [43].

Most of the rice heat tolerance genes identified by reverse genetics also play a positive regulatory role. The ectopic expression of *MSD1* maintains the normal grain filling and perfect grain production of rice under HS [44]. Qiao et al. isolated and cloned a member of the rice annexin family, *OsANN1*, which has calcium binding ability and ATPase activity, and endows abiotic stress tolerance by regulating the accumulation of antioxidants [45]. *SNAC3* encodes a stress-responsive NAC transcription factor that regulates the dynamic balance of H_2_O_2_ by regulating the expression of the ROS gene, thus achieving heat tolerance [46]. *OsNTL3* also encodes a NAC transcription factor that plays an important role in heat tolerance by transmitting HS signals/effects from the PM to the nucleus [47]. *OsHIRP1* encodes a RING finger E3 ligase which plays an active regulatory role in plant responses to HS, thus providing important information for plant adaptation and regulation under HS [48]. El-Esawi et al. cloned the *OsRab7* gene. *OsRab7* can improve the yield, drought resistance, and heat tolerance of transgenic rice by regulating the expression of osmotic regulators, antioxidants, and abiotic stress response genes [49]. *RGB1* encodes a rice heterotrimer G protein β subunit. Overexpression of *RGB1* in the rice seedling stage improves rice tolerance to HS, salt stress, and compound stress [50]. *OsCNGC14* and *OsCNGC16* are two cyclic nucleotide-gated ion channel proteins. *OsCNGC14* and *OsCNGC16 *are necessary for heat and cold tolerance and are regulators of calcium signal responses to temperature stress [51]. *OsNSUN2* encodes an RNA5-methylcytosine (M^5^C) methyltransferase in rice. The OsNUSN2-dependent mRNA M5C modification plays an important role in maintaining chloroplast function during HS [52]. Rice overexpressing *DPB3-1* showed stronger tolerance to HS without a negative effect on plant growth and yield [53].

### 3.2. Functional Genes That Play a Reverse Regulatory Role

Some genes negatively regulate the heat tolerance of rice. The knockout or reduction of the expression of these genes can significantly enhance the tolerance of rice to HS. Kan et al. identified a natural quantitative trait locus *TT2* by forward genetics, which encodes a G protein γ subunit and endows rice with heat tolerance in vegetative and reproductive stages without yield loss [54]. Liu et al. isolated a gene, namely *OsMDHAR4*, encoding monodehydroascorbate reductase from rice, which negatively regulates rice tolerance to HS by mediating stomatal closure induced by hydrogen peroxide [55]. *OsBN1* encodes rice fibrin, which plays an important role in the formation of plastids and lipid metabolism in chloroplasts, thus coordinating the regulation of rice growth and grain filling [56]. *OsUBP21* encodes a ubiquitin-specific proteolytic enzyme. OsUBP21-mediated protein deubiquitylation damages the regulation of basic heat tolerance in rice. The down-regulation of *OsUBP21* expression improves rice tolerance to HS [57]. The nitrate transporter *OsNRT2.3* is related to nitrogen use efficiency and the yield of rice. Allelic variation in the 5′ untranslated region of the *OsNRT2.3* gene leads to an increase in *OsNRT2.3b* protein expression during HS, thereby increasing nitrogen uptake efficiency in rice plants [58].

**Table 1 ijms-24-07140-t001:** Functional genes related to heat tolerance in rice cloned in recent years.

Gene	Gene Characteristics	Mechanism	Subcellular Localization	Expression Pattern	Function (Research) Period	Regulative Effect *	Reference
*TT1*	α2 subunit of the 26S proteasome	The degradation of ubiquitinated proteins			Seedling stage, flowering stage, and filling stage	+	[34]
*ER (ERECTA)*	Receptor-like kinase	Confers thermotolerance independent of water loss			Seedling and flowering stage	+	[35]
*OsHTAS*	RING finger ubiquitin E3 ligase	Through modulating ROS homeostasis to regulate stomatal aperture status	Nucleus and cytoplasm	All tissues surveyed and peaked in leaf blade	Seedling stage	+	[37]
*TOGR1*	DEAD box RNA helicase	Maintains pre-rRNA homeostasis under high temperatures by securing a proper pre-rRNA structure by elevating its helicase activity	Ducleolus	Regulated by both temperature and the circadian clock	Seedling stage	+	[38]
*Sus3*	Sucrose synthase	The increase in Sus3 expression leads to tolerance of high temperatures			Ripening stage	+	[39]
*SLG1*	Cytosolic tRNA 2-thiolation protein 2 (RCTU2)	Plays a key role in the response of rice plants to high temperature stress	Nucleus and cytoplasm	Universal expression	seedling and reproductive stages	+	[40]
*HTS1*	β-ketoacyl carrider protein re-reductase	Via the regulation of fatty acid biosynthesis and stress signaling	Thylakoid membrane	Predominantly expressed in green tissues and strongly induced by HS	Seedling stage	+	[41]
*TT3*	E3 ligase, chloroplast; and precursor protein	Protects chloroplasts to enhance thermotolerance	PM, endosomes, and chloroplast		Seedling stage, heading stage, and filling stage	+	[42]
*HTH5*	Pyridoxal phosphate homeostasis protein (PLPHP)	Reduces reactive oxygen species accumulation by increasing the heat-induced pyridoxal 5′-phosphate (PLP) content	Mitochondrion	Widely expressed	Heading stage	+	[43]
*MSD1*	Golgi/plastid-type manganese superoxide dismutase	Induced the expression of ROS scavengers, molecular chaperones, and the quality control system in developing seeds	Golgi apparatus and plastids	Actively expressed throughout the rice plant	Heading stage and filling stage	+	[44]
*OsANN1*	Rice annexin	By modulating the production of H_2_O_2_	Cell periphery and cytosol	Highly expressed in seeds and panicles	Seedling stage	+	[45]
*SNAC3*	NAC transcription factor	Through modulation of reactive oxygen species	Nucleus	Expressed ubiquitously	Seedling stage	+	[46]
*OsNTL3*	Membrane-associated NAC transcription factor	Through relaying HS signals/effects from PM to nucleus	PM and nuclues		Seedling stage	+	[47]
*OsHIRP1*	Heat-induced RING finger protein	OsHIRP1 is an E3 ligase that acts as a positive regulator in the plant response to HS	Cytoplasm and nucleus	Highly expressed under HS conditions	Germination stage	+	[48]
*OsRab7*	Small GTP-binding protein	By modulating osmolytes, antioxidants, and abiotic stress responsive genes expression			Seedling stage	+	[49]
*OsRGB1*	Heterotrimeric G protein beta subunit	Overexpression of OsRGB1 confers HS tolerance in rice			Germination and seedling stage	+	[50]
*Os-CNGC14*, *OsCNGC16*	Cyclic nucleotide-gated ion channel protein	The modulators of calcium signals in response to temperature stress	PM	Expressed in most organs	Seedling stage	+	[51]
*OsNSUN2*	RNA 5-methylatesine (M^5^C) me-methyltransferase	Plays essential roles in the maintenance of chloroplast function during heat acclimation	Nucleus	The highest expression level at the shoot tip	Seedling stage	+	[52]
*DPB3-1*	transcriptional regulator DNA polymerase II subunit B3-1	Increase HS tolerance in crops without negative effects on vegetative and reproductive growth				+	[53]
*TT2*	Gγ subunit	Through SCT1-dependent alteration of wax biosynthesis	Nucleus		Vegetative and reproductive growth period	−	[54]
*Os-MDHAR4*	Monodehydroascorbate reductase (MDHAR)	By mediating H_2_O_2_-induced stomatal closure	Chloroplasts	Expressed in all tissues surveyed and peaked in leaf blade	Seedling stage	−	[55]
*OsFBN1*	Fibrillin	Plays essential roles in plastoglobule formation and lipid metabolism in chloroplasts	Chloroplasts	highly expressed in green tissues	Seedling stage and reproductive growth stage	−	[56]
*OsUBP21*	Ubiquitin-specific protease	mediated protein de-ubiquitination plays a negative role in regulating basal thermotoleranance in rice	Intracellular	Mainly expressed in inflorescences, pistils, embryos, and shoots	Seedling stage	−	[57]
*OsNRT2.3*	Nitrate transporter	Required to maintain high yield and high nitrogen use efficiency				−	[58]

* +: positive regulation; −: negative regulation.

## 4. The Molecular Mechanism of The Rice Response to HS

In recent years, the molecular mechanisms mediated by heat-tolerant genes in rice have focused on maintaining the stability and fluidity of the PM, maintaining the dynamic balance of proteins, maintaining intracellular ROS homeostasis, and maintaining chloroplast stability and normal photosynthesis (Figure 1).

### 4.1. Stability and Fluidity of PM

PM is the first responder to HS. Indeed, the main response to HS also occurs in the PM, which is usually followed by changes in membrane fluidity and the activation of channels or receptors [59,60,61]. PM plays a key role in thermal sensing, the cellular response, and calcium signal transduction in plants [62]. High temperatures affect the fluidity and stability of the PM of activated membrane-related sensors (including calcium channels), resulting in a rapid influx of calcium ions, which in turn triggers a signal transduction cascade reaction [11,63,64]. Plant cells that survive at extreme temperatures first need to maintain the stability and fluidity of lipid membranes by regulating lipid saturation. Fatty acids constitute the basic component of membrane lipids. *HTS1* is a key factor in de novo fatty acid synthesis. *HTS1* deficiency directly damages fatty acid biosynthesis and the fat metabolism of *hts1* mutants. The decrease in fatty acid content destroys the integrity and stability of the cell membrane system under HS, resulting in an abnormal heat-induced calcium signal cascade [41]. *ER* improves the heat tolerance of rice through unknown signal transduction pathways and plays an important role in protecting plant cells from heat-induced cell injury and death [35]. *OsCNGC14* and *OsCNGC16* are located in the PM and are regulators of calcium signals in response to temperature stress, giving rice heat tolerance [51].

### 4.2. The Dynamic Balance of Protein

In addition to the stability of the PM, another major consequence of thermal damage is the disruption of protein homeostasis, which can lead to cell death and cytotoxicity. Under severe HS, a large number of proteins accumulate rapidly in cells in a short period, and removing these toxic proteins is more critical than restoring their activity [34]. The ubiquitin/26S proteasome system is an important protein degradation complex which is responsible for the degradation of ubiquitin-binding proteins. *TT1* encodes the 26S proteasome α2 subunit which effectively eliminates the cytotoxic denatured proteins involved in ubiquitination and maintains a dynamic protein balance under HS, thus protecting cells from heat damage [34]. RING finger ubiquitin E3 ligases *OsHIRP1* and *OsHTAS* play an active regulatory role in rice response to HS, and they may play a role in identifying and generalizing the target protein for subsequent degradation of 26S proteasome [37,48]. The expression of *OsHIRP1*, *OsHTAS*, and *TT1* was induced by high temperatures and the overexpression of any of these genes could improve the heat tolerance of rice [34,37,48]. HS upregulated the transcription level of ubiquitin-specific protein hydrolase *OsUBP21*. OsUBP21-mediated protein deubiquitylation harmed the regulation of basic heat tolerance in rice, and down-regulation of *OsUBP21* expression could improve rice tolerance to HS [57]. Protein dynamic balance under HS is closely related to translation regulation, which is related to the normal functions of mRNA, tRNA, and rRNA. DEAD box RNA helicase *TOGR1* endows rice with tolerance to HS by promoting its helicase activity under high temperatures, thus ensuring the stability of the pre-rRNA structure. The overexpression of *TOGR1* can protect rice growth at high temperatures [38]. The dynamic balance of tRNA is also very important to translation. *SLG1* encodes rice cytoplasmic tRNA2-thioprotein 2 which actively regulates the heat tolerance of rice by maintaining normal levels of vulcanized tRNA under HS, while the extensive effects of tRNA sulfation defects on protein homeostasis may lead to chronic protein toxicity stress, resulting in plant sensitivity to temperature [40]. *OsNSUN2* encodes M5C methyltransferase, which is necessary for the heat-triggered M5C modification of mRNAs involved in detoxification and photosynthesis. The dysfunction of *OsNSUN2* reduces the accumulation of proteins related to photosynthesis and the detoxification system under HS and leads to severe heat sensitivity [52].

### 4.3. The Accumulation of ROS

When plants continue to suffer from severe HS, the level of intracellular ROS increases significantly, thus breaking the dynamic balance of ROS and resulting in oxidative damage [45,65,66]. Excessive ROS will further destroy the structure and function of the biofilm, aggravate the lipid peroxidation and protein oxidation of the biofilm, increase the content of intracellular malondialdehyde (MDA), damage the normal function of proteins and nucleic acids, and even trigger programmed cell death [10,67,68]. High temperatures also impaired the activities of antioxidant enzymes, especially superoxide dismutase (SOD) and catalase (CAT) [65,69]. *OsANN1* enhances HS tolerance by regulating the accumulation of antioxidants under HS. The increased expression of *OsANN1* promotes the upregulation of SOD and CAT expression, thus clearing ROS as a stress defense mechanism [45]. Under HS, the structural high expression of *MSD1* immediately transformed O_2_− into H_2_O_2_ and significantly induced the expression of ROS scavengers, molecular chaperones, and quality control systems during seed development [44]. *SNAC3* maintains ROS homeostasis by directly activating genes encoding ROS scavengers and endows rice with heat tolerance [46]. The overexpression of *OsRab7* can reduce oxidative damage by inducing ROS-scavenging pathways and proteins involved in the defense mechanism. *OsRab7* can improve the yield and heat tolerance of transgenic rice by regulating the expression of antioxidants and abiotic stress response genes [49]. *HTS1* defects lead to abnormal heat-induced ROS signal transduction cascades, increased heat-induced H_2_O_2_ production, and decreased H_2_O_2_ scavenging ability, resulting in H_2_O_2_ accumulation and cell death [41]. ROS is also a signaling molecule that regulates processes including pathogen defense, programmed cell death, and stomatal behavior [70]. *OsHTAS* changes the stomatal state of leaves and enhances the heat tolerance of rice by mediating the accumulation of H_2_O_2_ [37]. Monodehydroascorbate reductase (MDHAR) encoded by *OsMDHAR4* is also a kind of ROS scavenger. *OsMDHAR4* negatively regulates rice tolerance to HS by mediating the stomatal state induced by H_2_O_2_. The inhibition of *Os MDHAR4* promotes stomatal closure and H_2_O_2_ accumulation, decreases water loss rate, and improves heat tolerance [55]. Mitochondria have also been proven to be the main producing and targeting sites of ROS [71]. The PLPHP protein encoded by *HTH5* is located in mitochondria and *HTH5* may reduce the damage to mitochondrial biological energy metabolism under HS by regulating the dynamic balance of ROS. Overexpression of *HTH5* can significantly reduce the ROS accumulation induced by HS [43].

### 4.4. Photosynthesis and Chloroplast Stability

Photosynthesis is one of the key physiological phenomena of plants affected by HS, and photosynthesis is very sensitive to high temperatures [72,73]. HS can destroy the membrane permeability of thylakoids and even disintegrate thylakoid grains, causing a decrease in chlorophyll content and resulting in changes in photochemical reactions and a decrease in the photosynthetic rate [69,74,75]. Chloroplasts are vulnerable to HS and when the temperature exceeds the normal tolerance range of crops, the yield is reduced. There are a large number of chloroplasts in mesophyll cells [76,77]. OsMDHAR4 was localized in chloroplasts, *OsMDHAR4* was expressed in all tested tissues, and the expression was the strongest in leaves, in which a large number of mesophyll cells were distributed, while the expression in roots was the lowest (almost no chloroplasts). *OsMDHAR4* deficiency led to a decrease in the water loss rate and an improvement in heat tolerance [55]. *OsFBN1* plays an important role in the lipid metabolism of chloroplasts. Rice fibrillin *OsFBN1* regulates the stability of the thylakoid membrane by affecting co-expression genes in photosynthesis, and finally affecting rice growth and grain filling under HS [56]. *OsNSUN2* is a RNA5-methyl cytosine (M5C) methyl transferase. HS enhances OsNSUN2-dependent M5C modification of mRNA involved in photosynthesis and the detoxification system, thus increasing protein synthesis. The heat-induced M5C modification plays an important role in maintaining chloroplast function and cell detoxification under HS [50]. Abnormal photosynthetic mechanisms and leaf senescence will affect photosynthesis [10]. HTS1 contains a chloroplast transport sequence which plays a key role in the chloroplast integrity of rice. *HTS1* encodes a β-ketoacyl carrier protein reductase (KAR) located in the chloroplast thylakoid membrane which is mainly expressed in green tissue, while the *hts1* mutant shows stronger chloroplast damage under HS [41]. In addition, *HTS1* also plays a role in rice leaf senescence [78]. Photosystem II (PSII) is the most sensitive component in photosynthetic devices and its activity is greatly affected by HS: it can even be partially terminated [73,79]. Less accumulation of mature *TT3.2* in chloroplasts is very important to protect thylakoids from HS. *TT3.2* is a chloroplast precursor protein and its accumulation is harmful to PSII complexes and thylakoids and can trigger chloroplast damage under HS. *TT3.1* maintains chloroplast stability by mediating *TT3.2* degradation [42].

## 5. Ways to Improve The Heat Tolerance of Rice

### 5.1. Agronomic Management

To cope with HS, most agronomic management measures mainly focus on the early sowing of rice, the adjustment of planting and irrigation systems, and the selection of early or late-maturing varieties to avoid high temperatures during grain filling [80,81]. To alleviate the yield loss of rice under HS during the reproductive period, the application rate of nitrogen fertilizer should be increased appropriately, and biochar and phosphorus fertilizer should be applied together [82,83]. Spray treatment at the flowering stage can rapidly reduce field temperature, delay leaf senescence, and increase the activity of antioxidant enzymes, thus reducing the yield loss of rice caused by HS [84]. In addition, the rational use of growth regulators such as CTK, BR, and ethylene precursors can reduce rice injury under HS [85,86,87,88,89]. Plant antioxidants, osmotic protective agents, and polyamines can also reduce the damage caused by HS, for example, endogenous ascorbic acid can reduce ROS accumulation and maintain leaf function [90].

### 5.2. Conventional Breeding

Improving the heat tolerance of rice by conventional breeding is an effective way to reduce the negative effects of HS on rice yield and quality. Conventional rice breeding was usually conducted based on phenotypes related to heat tolerance and applied in areas with a climate similar to that of rice-growing areas [17]. Therefore, the accurate evaluation of rice heat tolerance, the breeding of excellent varieties or lines, and the successful transfer of heat tolerance to specific varieties with good characteristics are of great significance for conventional breeding of rice heat tolerance. The commonly used evaluation indexes of rice heat tolerance are spikelet fertility, seed setting rate, yield per plant, root length, shoot length, pollen shedding level, heading date, culm length, panicle length, number of effective tillers, 1000-grain weight, and content of chlorophyll (Appendix A). Among them, spikelet fertility or seed setting rate is a typical index of heat tolerance of rice: it is direct, simple, and reliable and furthermore, it is the main index of conventional breeding. Using this index, a series of heat-tolerant rice materialswere identified, including Yuanjiang common wild rice [23,24,25], N22 [19,20,26,27], Giza178 [18], and Habataki [28], which can be used in conventional breeding to cultivate heat-tolerant rice varieties.

### 5.3. Molecular Marker-Assisted Breeding

As there are few strategies for agronomic management and conventional breeding, breeders must mine heat-resistant QTL and genes and apply them to rice breeding. At present, many heat-tolerant QTLs at different developmental stages of rice, such as the seedling stage, booting stage, and flowering stage, have been identified and verified (Appendix A). Using molecular markers linked to QTL, the identified QTL can be introduced into recipient varieties even if the potential gene is unknown. For example, using the molecular markers RM11633 and RM11642 linked to *qHTB1-1* to improve the heat tolerance of rice through molecular marker-assisted selection at the booting stage [25]. The identification and verification of heat-tolerant QTL with stable effects in different genetic backgrounds and different ecological environments and the polymerization of these non-allelic QTL was the goal of rice heat-tolerant molecular breeding [91].

### 5.4. Transgenic Methods and Genome Editing Technology

Genetic engineering is an efficient and time-saving method to cultivate heat-tolerant rice varieties [17,92]. *HTH5* is a HS resistance gene at the heading stage in rice. Under HS at the heading stage, the seed-setting rate of *HTH5* in the japonica genetic background of Sasanishiki increased by about 30%. The transgenic lines overexpressing *HTH5* not only had stable heat tolerance under indoor and field conditions but also did not affect other yield-related traits [43]. The overexpression of genes *TT1* [34], *ER* [35], *OsHTAS* [37], *TOGR1* [38], *Sus3* [39], *SLG1* [40], *HTS1* [41], *TT3* [42], *MSD1* [44], *OsANN1* [45], *SNAC3* [46], *OsNTL3* [47], *OsHIRP1* [48], *OsRab7* [49], *OsRGB1* [50], *OsNSUN2* [52], *DPB3-1* [53], *OsCNGC14* and *OsCNGC16* [51] could improve the heat tolerance of rice (Table 1). The heat tolerance of rice can also be improved by RNA interference or knockout of genes *TT2* [54], *OsMDHAR4* [55], *OsFBN1* [56], *OsUBP21* [57], and *OsNRT2.3* [58] (Table 1). Although transgenic rice can improve heat tolerance, the use of transgenic rice is still strongly affected due to public concerns about health and safety, which hinders the application of transgenic technology in practical breeding [93]. The emergence of genome editing technology brings new hope for the application of rice molecular breeding. It can breed plants with only target gene mutations without expression cassettes [94]. At present, genome editing has been successfully applied to the identification of heat-tolerant genes in rice and the study of molecular mechanisms.

## 6. Prospect

HS limits the overall growth and development of rice, resulting in a decrease in rice yield and quality, especially in the reproductive growth stage. The main effects of HS on rice include the destruction of protein homeostasis, damage to the photosynthetic mechanism, oxidative damage, and membrane instability. To reduce or avoid the loss of rice production caused by HS, it is necessary to clarify agronomic management measures, select tolerant varieties, and cultivate excellent rice lines. Therefore, there is an urgent need to identify and clone more heat-tolerant genes in rice and clarify the physiological and genetic mechanism of heat tolerance in rice and cultivate heat-tolerant varieties to improve the quality, yield, and tolerance of rice to HS. Accompanied with the emerging molecular evidence for rice heat tolerance QTLs and/or functional genes, it has been proved that genomic selecion and genome-editing technologies have higher selection efficiency and accuracy than traditional breeding method. However, it should be noted that the yield or quality characteristics of rice should not be affected by improving heat tolerance. In addition, artificial domestication will inevitably lead to a decrease in genetic diversity, and many heat-resistant traits may have been lost due to the strong selection of priority traits such as crop yield. Therefore, the identification and utilization of excellent natural alleles from wild rice and local varieties were an effective way for the heat-tolerant breeding of rice. It can be predicted that studies of rice heat tolerance QTLs and/or genes will broaden our knowledge about generating new breeding methods and improving breeding efficiency, and solve the main problems of heat tolerance in rice progressively.

## Figures and Tables

**Figure 1 ijms-24-07140-f001:**
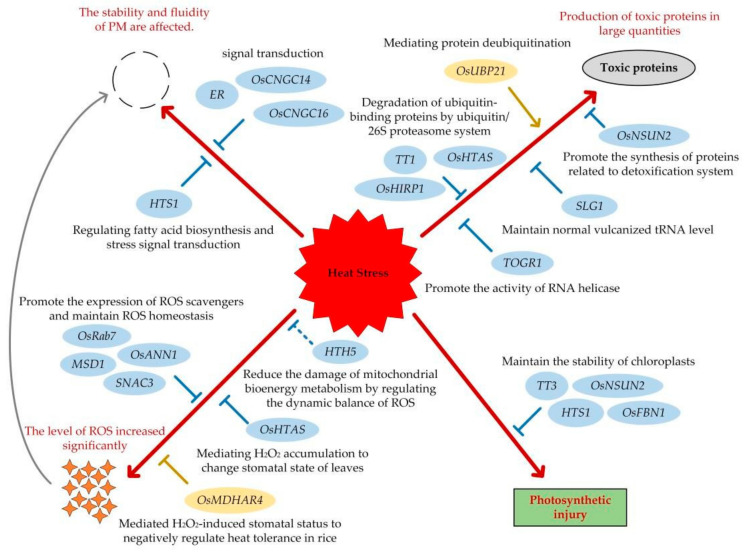
Molecular mechanism mediated by heat-tolerant genes in rice. HS mainly caused four major hazards to rice: the stability and fluidity of PM were affected, a large number of toxic proteins were produced, the level of ROS was significantly increased, and photosynthesis was damaged. Indeed, excessive ROS could also destroy the structure and function of the PM. *ER*, *OsCNGC14*, and *OsNGC16* improve the heat tolerance of rice through the signal transduction pathway. *HTS1* enhances heat tolerance by regulating fatty acid biosynthesis and stress signal transduction. *HTS1*, *TT3*, *OsFBN1*, and *OsNSUN2* improve heat tolerance by maintaining chloroplast stability. *OsNSUN2*, *TOGR1*, and *SLG1* improve heat tolerance by promoting the synthesis of proteins related to the interpretation system, promoting RNA helicase activity, and maintaining normal vulcanized tRNA levels, respectively. *OsUBP21* negatively regulates heat tolerance by mediating protein deubiquitylation, while *TT1*, *OsHIRP1*, and *OsHTAS* degrade ubiquitin-binding proteins through the ubiquitin/26S proteasome system. *OsHTAS* and *OsMDHAR4* regulate heat tolerance by mediating the stomatal state induced by H_2_O_2_. *HTH5* may reduce the damage to mitochondrial metabolism under HS by regulating ROS homeostasis. *OsANN1*, *MSD1*, *SNAC3*, and *OsRab7* maintain ROS homeostasis by promoting the expression of ROS scavengers.

## Data Availability

Data are contained within the article and Appendix A.

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
