# Peer review of "Genetic Research Progress: Heat Tolerance in Rice"

_ijms, 2023, doi:10.3390/ijms24087140_

Round 1

Reviewer 1 Report

Dear Authors,

I recommend this manuscript be continued for revisions. Here are some suggestions that we can deliver to make the article kind. Good luck.

Thank you.

Author Response

Dear Reviewer:

Thank you for your letter and for the reviewers’ comments concerning our manuscript entitled “Genetic Research Progress of Heat tolerance in Rice"(ID:ijms-2301412). Those comments are all valuable and very helpful for revising and improving our paper, as well as the important guiding significance to our researches. We have studied comments carefully and have made correction which we hope meet with approval. Revised portions are marked  in the paper.

Reviewer 2 Report

Manuscript: ijms-2301412

The authors of the review manuscript titled "Genetic research progress of heat tolerance in rice" have done a good job summarizing the recent findings of QTL and genes contributing to heat tolerance in rice. This review will be a useful resource for researchers in this field. However, the introduction section could benefit from additional information about the effects of heat stress on plants, particularly rice, to provide readers with a better understanding of the topic.

I also have a few minor suggestions and comments for the authors:

1.     It would be helpful to improve the written language throughout the manuscript. Although I understand the authors' message, there are some instances where the language is awkward, such as in the abstract's opening sentence where the term "national" is unclear. When I read this sentence, I started to wonder which country the authors are talking about when they said “national” food security. And right after this sentence, the authors wrote, “As an important food crop in the world”, I found that the authors probably wanted to say “international” food security if they care so much about the “world”? Improving the written language can enhance the manuscript's overall quality.

2.     I noticed that the authors' use of numbers and units could be more consistent with standard English writing. For example, while they correctly included spaces between numbers and units in some cases, such as "2-3 ℃" and "30-50 years," they did not do so consistently, such as in "304.2kb." Additionally, the authors should consider using a standard font for special characters like "℃," and they should include commas to break up long numbers, such as "3,555,000."

3.     I noticed that the acronym for "m5C" appeared twice in different formats (Lines 203 and 317). The authors should ensure that they use the correct format consistently.

Thank you for your attention to these suggestions, and I look forward to seeing the revised version of this manuscript.

Author Response

(The authors gave the same response as above.)

Round 2

Reviewer 1 Report

Dear Authors,

I recommend that this manuscript be continued for revision.

Heed the previous suggestions on 10 March.

Please reply to the questions in the comments box and revise the manuscript according to the unanswered suggestions.

I've also added some grammar suggestions to this script.

Thank You

Author Response

Dear Editors and Reviewers,

Thank you for your letter and comments on our manuscript titled“Genetic Research Progress of Heat tolerance in Rice" (ijms-2301412-review).These comments helped us improve our manuscript. We have addressed the reviewers' comments to the best of our abilities. We hope this meets your requirements for a publication. We marked the revised portions in red or blue in the manuscript. And an unmarked copy (clean version) of the revised manuscript also was uploaded. The main comments and our specific responses are detailed below:

The authors of the review manuscript titled "Genetic research progress of heat tolerance in rice" have done a good job summarizing the recent findings of QTL and genes contributing to heat tolerance in rice. This review will be a useful resource for researchers in this field. However, the introduction section could benefit from additional information about the effects of heat stress on plants, particularly rice, to provide readers with a better understanding of the topic.

Response 1:Thanks for the valuable comment, and we have added additional information about the effects of heat stress in reproductive growth period on rice as shown in introduction section in p1 line:38-45(The revised portions were marked in blue).

Two cited papers(Satake T. et al.1978; Zhu C.L. et al.2005) were added in references section in p13 line 30-32, one cited paper(Zhu S. et al.2017) was transferred to p13 line 28-29 from p14 line 10-12, and all the references were reordered.

I also have a few minor suggestions and comments for the authors:

  1. It would be helpful to improve the written language throughout the manuscript. Although I understand the authors' message, there are some instances where the language is awkward, such as in the abstract's opening sentence where the term "national" is unclear. When I read this sentence, I started to wonder which country the authors are talking about when they said “national” food security. And right after this sentence, the authors wrote, “As an important food crop in the world”, I found that the authors probably wanted to say “international” food security if they care so much about the “world”? Improving the written language can enhance the manuscript's overall quality.

Response 2:We corrected this kind of mistake to the best of our abilities in the article.

  1. I noticed that the authors' use of numbers and units could be more consistent with standard English writing. For example, while they correctly included spaces between numbers and units in some cases, such as "2-3 â„ƒ" and "30-50 years," they did not do so consistently, such as in "304.2kb." Additionally, the authors should consider using a standard font for special characters like "℃," and they should include commas to break up long numbers, such as "3,555,000."

Response 3:We corrected this kind of mistaketo the best of our abilities in the article.

  1. I noticed that the acronym for "m5C" appeared twice in different formats (Lines 203 and 317). The authors should ensure that they use the correct format consistently.

 Answer: Sorry for the mistake, and we corrected "m5C" to "M5C" in p9 line 34.

Round 3

Reviewer 1 Report

Dear Authors,

Thank you for the comments.

Regards.